# Soft- and Hard-Tissue Thicknesses in Patients with Different Vertical Facial Patterns and the Transverse Deficiencies, An Integrated CBCT-3D Digital Model Analysis

**DOI:** 10.3390/jcm12041383

**Published:** 2023-02-09

**Authors:** Alejandro Zaragoza Ballester, Álvaro Ferrando Cascales, José María Barrera Mora, Itamar Friedlander, Rubén Agustín-Panadero, Raúl Ferrando Cascales

**Affiliations:** 1Health Sciences PhD Program, San Antonio de Murcia Catholic University (UCAM), Los Jerónimos, nº 35, Guadalupe, 30107 Murcia, Spain; 2Department of Orthodontics, University of Sevilla, 41004 Sevilla, Spain; 3C/Gran de Gracia, nº 110, 08012 Barcelona, Spain; 4Prosthodontic and Occlusion Unit, Department of Stomatology, Faculty of Medicine and Dentistry, University of Valencia, C/Gascó Oliag 1, 46010 Valencia, Spain

**Keywords:** cone-beam computed tomography, transverse discrepancy, cortical thickness, facial type

## Abstract

Different vertical facial patterns may present different bone and gingival thicknesses at the molar level and can be influenced by the dental compensations that manifest in the presence of transverse bone discrepancies. A retrospective analysis was made of 120 patients divided into three groups according to their vertical facial patterns (mesofacial, dolichofacial or brachyfacial). Each group in turn was divided into two subgroups according to the presence or absence of transverse discrepancies assessed by cone-beam computed tomography (CBCT). The bone and gingival measurements were made integrating a CBCT-3D digital model of the patient dentition. In the brachyfacial patients, the distance from the palatine root to the cortical bone corresponding to the right upper first molar was significantly greater (1.27 mm) than in the dolichofacial (1.06 mm) and mesofacial (1.03 mm) (*p* < 0.05) patients. The brachyfacial and mesofacial patients with transverse discrepancies presented a greater distance from the mesiobuccal root of the left upper first molar and from the palatine root to the cortical bone, while in the dolichofacial individuals the distances were shorter (*p* < 0.05); The presence of transverse bone discrepancies in brachyfacial and mesofacial patients without posterior cross-bite implies a better dentoalveolar expansion prognosis than in dolichofacial individuals.

## 1. Introduction

The approach to orthodontic treatment and its goals may be modified by the different vertical facial patterns found in patients. There is a relationship between vertical morphological changes and breathing alterations, and the alveolar bone dimensions can also be expected to differ among the different vertical facial patterns [1,2].

Cortical bone tensions of variable intensity result in adaptations, and in this regard excessive tension levels stimulate increased bone formation, while low tensions induce bone loss [3]. The density and thickness of the cortical layer thus experience changes depending on the chewing force [4,5]. The hypodivergent population is characterized by a greater cortical thickness than hyperdivergent individuals at the level of the mandibular first and second molars [2,6,7].

Analyses have been made of cortical bone thickness in different facial patterns, relating it to the risk of fenestrations, dehiscences and even the success or failure of the use of microscrews. In this regard, significantly lesser both maxillary and mandibular bone thicknesses have been documented in dolichofacial individuals versus brachyfacial patients [8].

The incidence of gingival recession increases following orthodontic treatment, particularly in the presence of buccal inclination movements of the teeth; the existence of dentally masked transverse bone discrepancies, therefore, may pose an increased risk of periodontal damage [9].

Based only on cone-beam computed tomography (CBCT) scans, Germana et al. [10] developed a method for determining gingival thickness and the distance to the alveolar crest. Integration of CBCT and digital scanning of STL (STereoLithography) models allows us to perform measurements of the buccal and palatine/lingual gingival portion [11].

The WALA ridge can be used to determine the apical base of the mandibular bone; it is useful for designing an individualized dental arch shape and can be used in the maxillary arch as a reference for maxillary width [12,13,14]. According to Andrews, the WALA ridge coincides with the most prominent part of the buccal alveolar bone, and may be established as the limit of our dentoalveolar expansions [15].

Although a number of studies have examined the association between cortical bone thickness and vertical facial pattern [6,8,16,17], no analyses have been made relating it to the transverse bone dimensions. Cone-beam computed tomography-based studies likewise have not analyzed gingival thickness in various vertical facial patterns with and without transverse deficiency.

Thus, the present study was carried out to compare the distances to the WALA ridge and WALA bone from the buccal root surface, and from the palatine root surface of upper molars to the cortical bone and buccal and palatine gingival portion, in the different facial patterns, in the presence and absence of transverse bone discrepancies, and in the combination facial pattern—transverse bone discrepancy.

## 2. Materials and Methods

The patients included in this retrospective study were selected from the database of the Master of Orthodontics and Dentofacial Orthopedics of the Universidad Católica de Murcia (Murcia, Spain). The inclusion criteria were: (a) patients over 18 years of age; (b) permanent dentition; (c) presence of fully erupted upper and lower first and second molars; (d) no previous orthodontic treatment; (e) no posterior cross-bite; (f) no facial asymmetry defined as a chin deviation of over 3 mm in resting frontal photographic projection; and (g) no major restorations of upper and lower first and second molars. The study was approved by the Universidad Católica de Murcia.

To secure a statistical power of 80% with a level of significance of 5%, the minimum sample size was seen to be 120 cases (20 per combination of facial pattern and transverse bone discrepancy). With this sample size, multiple analyses of variance (MANOVA) comparing the linear measurements according to the combination of facial pattern and discrepancy would be able to detect statistically significant differences between groups with a 0.10 effect size. Sample size estimation was performed using the G*Power 3.1 application.

A sample of 120 individuals (45 males and 75 females) was selected (median age 25.5 years [range 20–33]). The included cases belonged to the patient database of the Master of Orthodontics of the Universidad Católica de Murcia (Spain). The database contained a total of 286 histories of patients seen in the Department of Orthodontics during the period between March 2018 and January 2022. Following an analysis of the necessary records and of compliance with the inclusion/exclusion criteria, a sample of 178 potentially recruitable patients was established, of which an analysis was made of transverse discrepancy in the CBCT study and of the facial pattern from lateral teleradiography of the skull, obtaining: 57 brachyfacial patterns (32 with transverse discrepancy and 25 without discrepancy) 66 mesofacial patterns (39 with transverse discrepancy and 27 without discrepancy) and 55 dolichofacial patterns (29 with transverse discrepancy and 26 without discrepancy). Random selection was then made (based on a computer-generated list) to establish the following 6 groups: (I) brachyfacial with transverse bone discrepancy (n = 20; 9 males and 11 females); (II) brachyfacial without transverse bone discrepancy (n = 20; 8 males and 12 females); (III) mesofacial with transverse bone discrepancy (n = 20; 8 males and 12 females); (IV) mesofacial without transverse bone discrepancy (n = 20; 9 males and 11 females); (V) dolichofacial with transverse bone discrepancy (n = 20; 7 males and 13 females); (VI) dolichofacial without transverse bone discrepancy (n = 20; 4 males and 16 females). The MANOVA models were adjusted a posteriori for sex and age. The distribution of the study sample is reflected in Table 1.

Records taken included CBCT, lateral cephalometry and a digital model (STL file) of the patient dentition. The facial pattern was determined using the VERT index of Ricketts’ cephalometry [18]. A VERT index of −0.5 to +0.5 was interpreted as indicating a mesofacial pattern, > +0.5 as indicating a brachyfacial pattern, and < −0.5 as indicating a dolichofacial pattern. For the analysis of transverse bone discrepancy, a CBCT-based analysis was performed following the method of the University of Pennsylvania [12] (Figure 1). Discrepancy was classified as “with transverse discrepancy” when the maxillary-mandibular difference (Mx-Md) was <5 mm, and as “without transverse discrepancy” in the case of discrepancy ≥ 5 mm.

All subjects had analogue plaster models made. These were subsequently scanned with a CS 3600 scanner (Carestream Dental^®^, Atlanta, GA, USA), with which the STL files were obtained. For CBCT acquisition, the Orthophos SL 2D/3D (Dentsply Sirona^®^, Charlotte, NC, USA) system was used. The images were acquired with settings of 85 Kvp and 10 mA, with an effective exposure time of 4.4 s. The volume area of the object/field of view (FoV) corresponded to 11 cm × 10 cm. The images were saved as DICOM files and subsequently used for analysis employing NemoStudio 2018 (Nemotec^®^, Madrid, Spain).

The different CBCT views were reoriented in the three spatial dimensional planes (Figure 2). In the sagittal view, the anatomical occlusal plane was aligned parallel to the sagittal reference plane which lies parallel to the ground. In the coronal view, the CBCT view was oriented parallel to the horizontal reference plane, which lies parallel to the ground. Finally, for axial orientation, the CBCT view was positioned by matching the anterior nasal spine (ANS) with a vertical line perpendicular to the horizontal reference plane.

The integration of the digital model (STL file) and the CBCT scan (DICOM files) was carried out with NemoStudio 2018 (Nemotec^®^, Madrid, Spain) using a dot plotting process. The different CBCT views (axial, coronal and sagittal), together with the three-dimensional (3D) volumetric reconstruction, were used for dot plotting, seeking the most reproducible anatomical areas in both files (CBCT and digital model). The plots were mainly located at the incisal edges, cuspids and bottoms of the main sulci, as the areas of best reproduction between the two. Subsequently, a second surface adjustment was made in order to reduce the margin of error. For this second adjustment, the anterior region of the maxilla and mandible was marked as a reference on the digital model (Figure 3).

For measurement of the distances in millimeters, we first traced a reference line joining WALA-WALA. The linear measurements were made on this line from the external root surface of the upper first and second molars to the buccal bone portion (WALA bone) and to the palatine portion. Similarly, on this same line, measurements were made to the buccal and palatine gingival portion. A total of 5 measurements were obtained for each first molar (three bone and two gingival measurements), while only the three bone measurements were obtained for the second molars (Figure 4).

### Statistical Analysis

A descriptive analysis was made of each variable, including the mean and standard deviations (SD). The differences between the linear measurements according to the facial pattern and the presence or absence of transverse maxillo-mandibular bone discrepancies were evaluated based on multivariate analyses of variance (MANOVA), grouping the most related parameters and applying MANOVA to each subgroup to control correlations between them, after confirming normal data distribution with the Shapiro-Wilks test. All the variables exhibited a normal distribution. Statistical significance was considered for *p* < 0.05. The statistical results were accompanied by the effect size assessed by Cohen’s d (small d = 0.2–0.3, medium d = 0.5–0.8 and large d = over 0.8). The analyses were performed using the SPSS version 25 statistical package for MS Windows (IBM Corp., Armonk, New York, NY, USA).

All the measurements were made by the same examiner. In order to determine the intra-examiner error, we randomly selected 60 subjects in which all the measurements were repeated during an interval of two weeks. The error was calculated based on the intraclass correlation coefficient (ICC).

## 3. Results

The ICC was over 0.98 (*p* < 0.001) for all the variables, with a mean percentage discrepancy of 1.06% (±0.77%)—indicating that the operator was consistent during repetition of the measurements.

In relation to the upper molars, correlations were established between the linear buccal bone, palatine-lingual and gingival measurements in the different facial patterns, in the presence and absence of transverse bone discrepancies, and the combination facial pattern—transverse bone discrepancy. The mean values and standard deviations of the MANOVA model and corresponding 95% confidence intervals (95%CIs) are reported in Table 2, Table 3, Table 4, Table 5 and Table 6.

In the case of the molars of the first quadrant (teeth 1.6 and 1.7), statistically significant differences were found only between the linear measurements and the different relations corresponding to the distance of tooth 16-osseous P with respect to the vertical facial pattern (F(2) = 3.804, sig. = 0.024), with higher mean values in patients with a brachyfacial pattern (M = 1.27, SD = 0.51) than in those with the rest of patterns (M = 1.06 mm, SD = 0.36 for dolichofacial individuals and M = 1.03 mm, SD = 0.36 for mesofacial profiles). Specifically, the mean difference between the brachyfacial and dolichofacial patterns was D = 0.207 mm, 95%CI [0.021, 0.435], with Cohen’s d = 0.485, while the difference between the brachyfacial and mesofacial patterns was D = 0.238 mm, 95%CI [0.010, 0.466], with Cohen’s d = 0.557 (Table 2).

In the case of the molars of the second quadrant (teeth 2.6 and 2.7), statistically significant differences were found for the distance 26-osseous MV with the combination facial pattern-discrepancy (F(2) = 3.127, sig. = 0.048). In the presence of transverse discrepancy, the mean values for mesofacial (M = 2.13 mm, SD = 1.17) and brachyfacial patterns (M = 2.46 mm, SD = 1.15) were greater than in the absence of transverse discrepancy (M = 1.53 mm, SD = 0.69 and M = 1.60 mm, SD = 0.94, respectively)—resulting in differences of D = 0.605 mm, 95%CI [0.079, 1.289], d = 0.541 and D = 0.856 mm, 95%CI [0.172, 1.540], d = 0.766, respectively (Figure 5) (Table 7).

In relation to the distances 26-osseous P and 26-gingival P, significant differences were recorded in the presence and absence of transverse bone discrepancy (F(1) = 10.767, sig. = 0.001 and F(1) = 22.692, sig. = 0.000, respectively). Specifically, in the presence of transverse discrepancy, both the mean of 26-osseous P (M = 2.62 mm, SD = 1.74) and of 26-gingival P (M = 4.99 mm, SD = 1.39) were greater than in the absence of transverse discrepancy (M = 1.72 mm, SD = 1.31 and M = 3.75 mm, SD = 1.42, respectively)—resulting in differences of D = 0.901 mm, 95%CI [0.357, 1.445], d = 0.564 for 26-osseous P and D = 1.240 mm, 95%CI [0.724, 1756], d = 0.810 for 26 gingival P (Table 5). Significant differences were found on analyzing the combination discrepancy-facial pattern for 26-osseous P (F(2) = 4.269, sig. = 0.016). Specifically, in the presence of transverse bone discrepancy, the means corresponding to the brachyfacial and mesofacial patterns (M = 2.67 mm, SD = 1.48 and M = 2.99 mm, SD = 1.74, respectively) were greater than in the absence of transverse discrepancy (M = 1.29 mm, SD = 0.48 and M = 1.43 mm, SD = 0.92, respectively), obtaining differences of D = 1.373 mm, 95%CI [0.431, 2.314], d = 0.736 for brachyfacial individuals, and D = 1.558 mm, 95%CI [0.617, 2.500], d = 0.781 for mesofacial patients (Figure 6) (Table 7).

## 4. Discussion

The design of the present study is based on a novel method for evaluating the distances from the upper first and second molars to the WALA ridge. This was done by CBCT-STL integration, which allowed analysis to the bone surface (WALA bone) and the gingival portion (WALA ridge) (Figure 4). The CBCT-STL integration method has already been used in the literature to measure bone and gingival thickness, but only at the upper incisor and canine level [11].

Our study sample consisted of 120 patients over 18 years of age and classified according to their vertical facial patterns and the presence or absence of transverse bone discrepancies, based on the transverse analytical method of the University of Pennsylvania [12]. Characterization of the pattern was based on the Vert index [18], which involves 5 factors determining the vertical facial pattern.

Using scanned plaster models and CBCT images, Timothy et al. [19] analyzed the location of the center of resistance of the lower molars with respect to the WALA ridge and its association to inclination, though the authors did not perform integration of the two types of files. In our series involving different vertical facial patterns with and without transverse bone discrepancy, we analyzed the distance from the WALA ridge to the root portion, over a line joining WALA-WALA (CBCT-STL integrating), following orientation of the CBCT scan. At the palatine level we also carried out measurements from the root structure to the gingival margin. In addition, linear measurements were made from the WALA bone to the root portion and from the palatine root to the cortical bone.

Determination of the basal bone level remains a confusing issue among the authors, as there is no single criterion. Andrews [15] proposed the WALA points to estimate the width of the basal arch, but this can be altered by soft tissue modifications. Al-Hilal et al. [20] in turn proposed a novel method for analyzing the mandibular basal width at the level of the canines and first molars that would not be affected by the soft tissues. They chose the junction of the middle third with the apical third at canine level as apical reference, and for the transverse measurements, selected the midpoint between the buccal and lingual cortical layers to perform the basal measures in the axial view. On comparing the mandibular basal width between males and females with class I and class II division 1 malocclusions, they recorded greater basal widths in males versus females, and the basal widths at canine level were moreover smaller in subjects with class II division 1 malocclusions than in those with class I malocclusions. Lastly, Nahas et al. [21], in their volumetric study of the maxilla and mandible, considered elimination of the maxillary and mandibular crowns for performing the bone volumetric measurements.

It would be logical to assume that dolichofacial individuals with thinner cortical bone [6,7,8,16] would present shorter distances to the WALA ridge and to the palatine cortical bone compared with the rest of the facial patterns. Alhawasli et al. [22], in their volumetric study of the maxilla and mandible, also found mandibular bone volume in hypodivergent class III individuals to be significantly greater than in hyperdivergent class III patients. Our data evidenced no significant differences in this regard, except for the distance from the palatine root of the right upper first molar to the cortical bone (16-osseous P), where the mean distance in the brachyfacial patients (1.27 mm) was greater than in the rest of the patterns (*p* < 0.05).

Sadek et al. [17] also analyzed the alveolar and skeletal dimensions among individuals with different vertical facial biotypes using CBCT. These authors performed measurements of alveolar thickness and height throughout the region of alveolar support of the teeth. They found the dolichofacial group to present greater anterior dentoalveolar height, with no significant differences in posterior alveolar height, in both the maxilla and the mandible. In addition, in relation to bone thickness, the dolichofacial group was characterized by thinner alveolar bone in the anterior region of the maxilla and at almost all sites in the mandible. These different bone thicknesses of the anterior maxillary region in the various vertical facial patterns may also be related to differences in inclination of the anterior maxillary alveolar process, since it has been seen that brachyfacial patterns are characterized by a more buccal inclination than in dolichofacial individuals [23]. Horner et al. [16] likewise measured buccal and lingual cortical bone thickness at 16 inter-radicular sites of the maxilla and the mandible. Hypodivergent patients presented significantly thicker buccal cortical bone. On the other hand, the lingual cortical bone of the maxilla was also significantly thicker in the hypodivergent subjects than in the hyperdivergent patients.

The different buccolingual inclinations of the molars can also affect the linear bone measurements, since when the molars that are excessively inclined buccally, the distance to the WALA ridge may decrease, in the same way as the distance to the palatine cortical bone. The literature does not clarify whether the different vertical facial patterns are characterized by differences in buccolingual inclination of the upper molars. Janson et al. [24] recorded greater buccal inclination of the maxillary first molars in dolichofacial patterns, in the same way as reported by Mitra [25] in relation to the second molars. However, Eraydin et al. [26] observed no significant differences between patterns. What we do know is that in the presence of transverse bone discrepancy, the upper molars respond with increased buccolingual inclination, compensating for the basal discrepancy problem [27].

We thus considered it necessary to analyze the transverse bone relationship in the different vertical facial patterns. Our results showed that in brachyfacial and mesofacial individuals with transverse bone discrepancy, the distance from the mesiobuccal root to WALA bone, and the distance from the palatine root to the cortical bone in the case of the upper left first molar, were significantly greater than in the absence of transverse bone discrepancy (Figure 4 and Figure 5). The opposite was observed in the dolichofacial patients, however. The greater 26-osseous P distance found on exclusively analyzing the transverse factor could be explained by the existence of greater distances in the presence of transverse bone discrepancies in the brachyfacial and mesofacial groups.

These observations are of clinical relevance, since they suggest that patients with transverse bone discrepancies and a mesofacial or brachyfacial pattern will have better prognoses than dolichofacial individuals, as the latter would have a more limited dental compensation capacity. This could imply an increased risk of bone dehiscence in dolichofacial patients in the event that dentoalveolar expansion is planned to solve a transverse bone discrepancy in an individual without posterior cross-bite [28].

The greater or lesser buccal inclination of the teeth could also be related to the gingival biotype or gingival thickness. In this respect, Zawawi et al. [29] found proinclined and protruded mandibular incisors to be associated with a thin gingival biotype.

Very few studies have examined the relationship between the gingival phenotype and the facial biotype. Moussa Assiri et al. [30] explored the relationship between the different gingival phenotypes (thin and thick) and the vertical facial patterns (mesofacial, dolichofacial and brachyfacial). These authors observed a significant association between the presence of a thin gingival biotype and the mesofacial pattern, and concluded that mesofacial individuals are more likely to have a thin gingival biotype.

In our study, CBCT-STL integration allowed us to perform measurements of the buccal and palatine gingival portion. The measurements obtained were compared among the different vertical facial patterns, and with the combination facial pattern-transverse bone discrepancy. No significant differences were recorded for any of the measurements (*p* > 0.05) among the different combinations analyzed (vertical pattern, presence of transverse bone discrepancy, vertical pattern-transverse bone discrepancy). Significant differences were found only for palatine gingival thickness in the upper left first molar (26-gingival P); accordingly, and without considering the vertical facial pattern, the patients with transverse bone discrepancy presented greater thickness values (4.99 mm versus 3.75 mm) (*p* < 0.05).

The present study has a number of limitations, such as the absence of inter-examiner calibration. On the other hand, we did not take the buccolingual inclination of the molars into account for their relationship with the linear measurements, and did not consider the sagittal relationship of the molars (classes I, II and III). Similarly, no distinctions in the results were made in relation to patient sex or gender. The location of the WALA ridge was taken to be the portion of maximum buccal convexity, and in cases where the latter was found to be flattened, the molar furcation zone was taken as a reference.

## 5. Conclusions

In brachyfacial individuals, the mean distance from the palatine root to the cortical bone in the case of the upper right first molar (1.6) was significantly greater than in the dolichofacial and mesofacial patients.

Brachyfacial and mesofacial individuals with transverse bone discrepancy presented a significantly greater distance from the mesiobuccal root of the upper left first molar (2.6) to the WALA bone and from the palatine root to the cortical bone, compared with dolichofacial individuals.

## Figures and Tables

**Figure 1 jcm-12-01383-f001:**
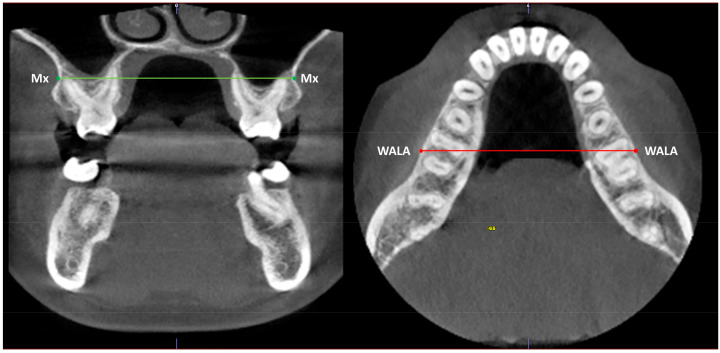
Maxillomandibular transverse skeletal discrepancy analysis.

**Figure 2 jcm-12-01383-f002:**
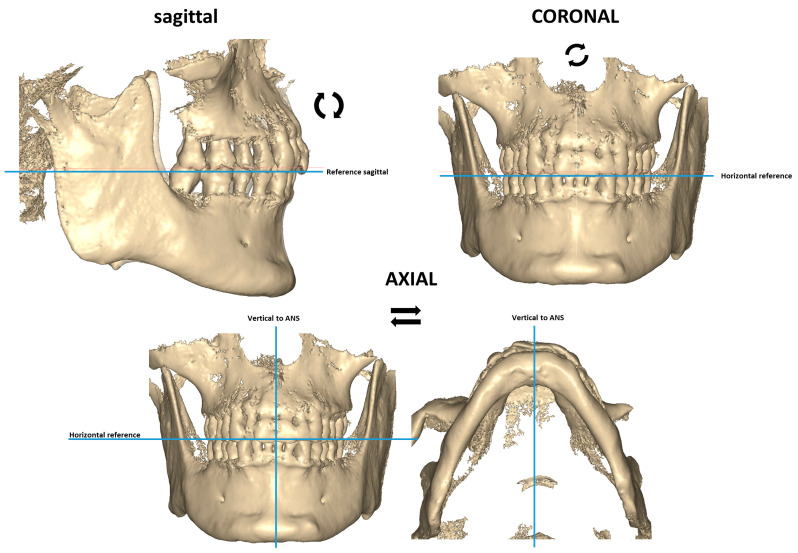
Orientation of the different CBCT views.

**Figure 3 jcm-12-01383-f003:**
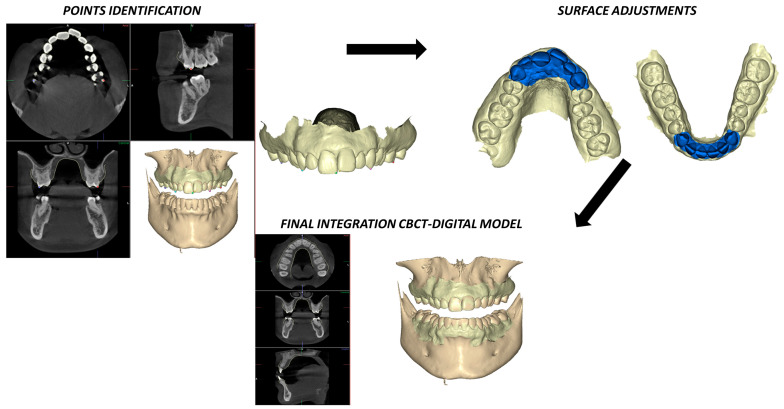
Cone-beam computed tomography overlay and digital model using dot plotting.

**Figure 4 jcm-12-01383-f004:**
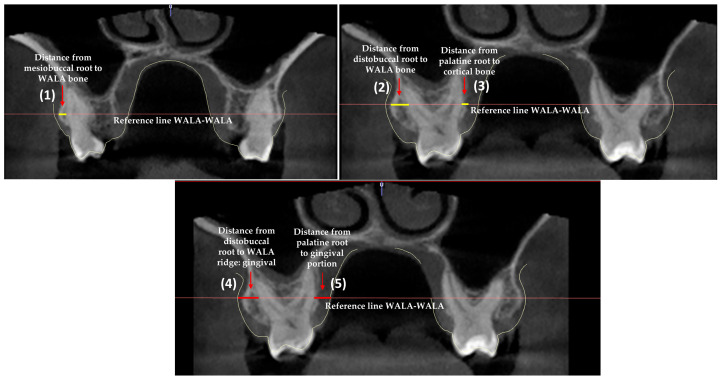
Representation of the 5 measurements made corresponding to each upper first molar. The measurements of the second upper molars were made in the same way but only for the bone variables. (1) Distance from mesiobuccal root to WALA bone: osseous V (MV). (2) Distance from distobuccal root to WALA bone: osseous V (DV). (3) Distance from palatine root to cortical bone: osseous P. (4) Distance from distobuccal root to WALA ridge: gingival (DV). (5) Distance from palatine root to gingival portion: gingival P.

**Figure 5 jcm-12-01383-f005:**
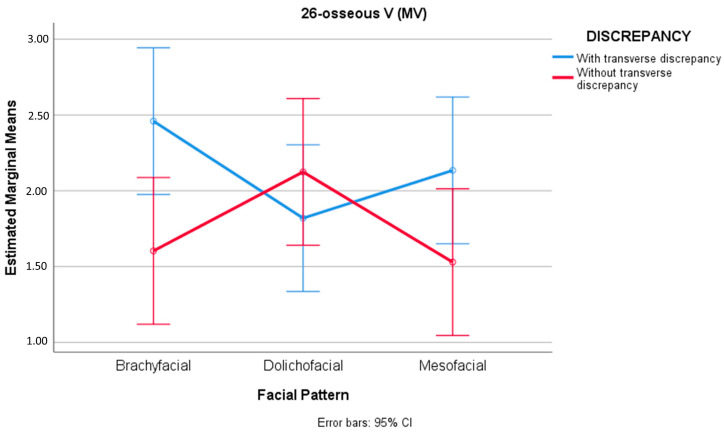
Transverse bone discrepancy inverts the behavior of the distance for the mesiobuccal root of the right upper first molar between patterns; in the absence of transverse bone discrepancy, the dolichofacial values tend to be greater than in the rest of the facial patterns, while in the presence of transverse bone discrepancy, the dolichofacial values are lower than in the rest of the patterns.

**Figure 6 jcm-12-01383-f006:**
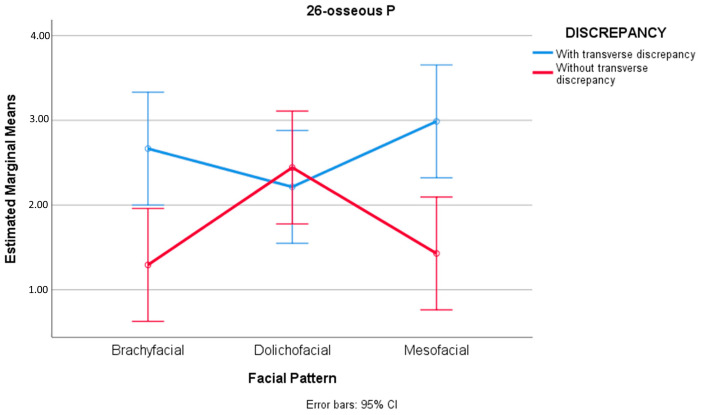
Transverse bone discrepancy inverts the behavior of the distance for the palatine root of the right upper first molar between patterns; in the absence of transverse bone discrepancy, the dolichofacial values tend to be greater than in the rest of the facial patterns, while in the presence of transverse bone discrepancy, the dolichofacial values are lower than in the rest of the patterns.

**Table 1 jcm-12-01383-t001:** Number of subjects and final distribution of the study sample.

	Total (n = 120)			Mesofacial (n = 40)			Dolichofacial (n = 40)			Brachyfacial (n = 40)			*p*-Value
Sex (%) Male/Female	37.5%/62.5%			42.5%/57.5%			27.5%/72.5%			42.5%/57.5%			0.278
Age (Median [IQR])	25.5 [20.0–33.0]			26.0 [19.5–32.5]			22.5 [18.0–33.5]			26.5 [21.0–32.5]			0.657
Maxilla (mean ± SD)	58.10 ± 3.23			57.82 ± 2.84			57.96 ± 3.32			58.52 ± 3.55			0.602
Mandible (mean ± SD)	54.20 ± 2.85			53.72 ± 2.72			54.41 ± 2.87			54.48 ± 2.95			0.425
Mx-Md (mean ± SD)	3.90 ± 3.13			4.10 ± 2.90			3.55 ± 3.26			4.04 ± 3.27			0.692
	With transverse maxillo-mandibular deficit (n = 60)	Without transverse maxillo-mandibular deficit (n = 60)	*p*-value	With transverse maxillo-mandibular deficit (n = 20)	Without transverse maxillo-mandibular deficit (n = 20)	*p*-value	With transverse maxillo-mandibular deficit (n = 20)	Without transverse maxillo-mandibular deficit (n = 20)	*p*-value	With transverse maxillo-mandibular deficit (n = 20)	Without transverse maxillo-mandibular deficit (n = 20)	*p*-value	
Sex (%) Male/Female	40.0%/60.0%	35.0%/65.0%	0.572	40%/60%	45%/55%	0.759	35%/65%	20%/80%	0.288	45%/55%	40%/60%	0.749	
Age (Median [IQR])	28.0 [20.5–35.0]	23.0 [19.0–30.0]	0.052	26.5 [21.0–34.0]	24.5 [18.5–29.5]	0.398	29.5 [19.5–38.5]	21.5 [18.0–26.5]	0.046 *	27.5 [20.5–32.0]	25.5 [21.0–32.5]	0.925	
Maxilla (mean ± SD)	56.42 ± 2.87	59.78 ± 2.67	0.000 **	56.49 ± 2.80	59.16 ± 2.21	0.002 **	56.41 ± 3.13	59.52 ± 2.78	0.002 **	56.36 ± 2.80	60.67 ± 2.87	0.002 **	
Mandible (mean ± SD)	55.14 ± 2.92	53.26 ± 2.45	0.000 **	54.82 ± 2.94	52.62 ± 2.00	0.009 **	55.56 ± 2.72	53.26 ± 2.60	0.002 **	55.04 ± 3.20	53.91 ± 2.64	0.234	
Mx-Md (mean ± SD)	1.28 ± 1.68	6.52 ± 1.72	0.000 **	1.66 ± 1.63	6.54 ± 1.45	0.000 **	0.84 ± 1.96	6.26 ± 1.58	0.000 **	45%/55%	6.76 ± 2.10	0.000 **	

*p*-value for age (Pearson Chi^2^); *p*-value for age (nonparametric Kruskal-Wallis test and Mann-Whitney U-test); *p*-value for maxilla, mandible and Mx-Md (parametric Student *t*-test and single-factor ANOVA); * Significant result 5% (*p* < 0.05); ** Significant result 1% (*p* < 0.01); (IQR) Interquartile range.

**Table 2 jcm-12-01383-t002:** Linear measurements corresponding to upper right molars in the different vertical facial patterns.

		Pattern				*p*-Value
		Total	Brachyfacial	Dolichofacial	Mesofacial	
16-OSSEOUS V (DV)	N	120	40	40	40	
	Mean	2.55	2.51	2.46	2.68	0.438
	SD	0.79	0.79	0.91	0.63	
	95%CI	(2.41, 2.69)	(2.26, 2.77)	(2.17, 2.75)	(2.48, 2.88)	
16-GINGIVAL (DV)	Mean	3.29	3.27	3.16	3.42	0.433
	SD	0.9	0.94	0.98	0.77	
	95%CI	(3.12, 3.45)	(2.97, 3.57)	(2.85, 3.48)	(3.18, 3.67)	
16-OSSEOUS V (MV)	Mean	1.48	1.57	1.36	1.53	0.286
	SD	0.63	0.63	0.67	0.59	
	95%CI	(1.37, 1.6)	(1.37, 1.77)	(1.14, 1.57)	(1.34, 1.72)	
16-OSSEOUS P	Mean	1.12	1.27	1.06	1.03	0.024 *
	SD	0.43	0.51	0.36	0.36	
	95%CI	(1.04, 1.2)	(1.1, 1.43)	(0.95, 1.18)	(0.91, 1.15)	
16-GINGIVAL P	Mean	4.05	3.82	4.34	4.01	0.063
	SD	1	0.96	0.98	1.02	
	95%CI	(3.87, 4.24)	(3.51, 4.12)	(4.02, 4.65)	(3.69, 4.34)	
17-OSSEOUS V (DV)	Mean	2.66	2.52	2.78	2.69	0.546
	SD	1.13	1.09	1.13	1.19	
	95%CI	(2.46, 2.87)	(2.17, 2.87)	(2.42, 3.14)	(2.31, 3.07)	
17-OSSEOUS P	Mean	1.69	1.75	1.8	1.53	0.405
	SD	0.93	0.92	1.06	0.78	
	95%CI	(1.52, 1.86)	(1.45, 2.04)	(1.46, 2.13)	(1.28, 1.78)	
17-OSSEOUS (MV)	Mean	2.45	2.44	2.38	2.54	0.792 (ANOVA)
	SD	1.01	1.04	0.91	1.1	
	95%CI	(2.27, 2.64)	(2.1, 2.77)	(2.09, 2.68)	(2.19, 2.89)	

Estimated marginal means, SD and 95%CI, significance between subjects effects test (MANOVA); * significant *p* ≤ 0.05, significant *p* ≤ 0.01. V: Buccal; P: Palatine; DV: Distobuccal; MV: Mesiobuccal.

**Table 3 jcm-12-01383-t003:** Linear measurements corresponding to upper left molars in the different vertical facial patterns.

		Pattern				*p*-Value
		Total	Brachyfacial	Dolichofacial	Mesofacial	
26-OSSEOUS V (DV)	Mean	2.53	2.49	2.6	2.49	0.802
	SD	0.83	0.89	0.93	0.66	
	95%CI	(2.38, 2.68)	(2.2, 2.78)	(2.3, 2.9)	(2.28, 2.7)	
26-GINGIVAL V (DV)	Mean	3.32	3.4	3.25	3.31	0.826
	SD	1.09	1.09	1.14	1.07	
	95%CI	(3.12, 3.52)	(3.05, 3.75)	(2.89, 3.61)	(2.97, 3.65)	
26-OSSEOUS V (MV)	Mean	1.95	2.03	1.97	1.83	0.718
	SD	1.12	1.12	1.24	1	
	95%CI	(1.74, 2.15)	(1.67, 2.39)	(1.58, 2.37)	(1.51, 2.15)	
26-OSSEOUS P	Mean	2.17	1.98	2.33	2.21	0.615
	SD	1.6	1.29	1.89	1.58	
	95%CI	(1.88, 2.46)	(1.57, 2.39)	(1.72, 2.93)	(1.7, 2.71)	
26-GINGIVAL P	Mean	4.37	4.32	4.49	4.3	0.836
	SD	1.53	1.38	1.62	1.61	
	95%CI	(4.09, 4.65)	(3.88, 4.76)	(3.97, 5.01)	(3.79, 4.82)	
27-OSSEOUS (DV)	Mean	2.59	2.43	2.76	2.57	0.393
	SD	1.08	0.93	1	1.28	
	95%CI	(2.39, 2.78)	(2.13, 2.73)	(2.44, 3.08)	(2.16, 2.98)	
27-OSSEOUS (MV)	Mean	2.29	2.3	2.34	2.23	0.852
	SD	0.92	1.04	0.79	0.94	
	95%CI	(2.12, 2.45)	(1.96, 2.63)	(2.09, 2.59)	(1.93, 2.53)	
27-OSSEOUS P	Mean	1.61	1.63	1.54	1.67	0.768
	SD	0.82	0.7	0.8	0.95	
	95%CI	(1.47, 1.76)	(1.41, 1.86)	(1.29, 1.79)	(1.37, 1.97)	

Estimated marginal means, SD and 95%CI, significance between subjects effects test (MANOVA); significant *p* ≤ 0.05, significant *p* ≤ 0.01. V: Buccal; P: Palatine; DV: Distobuccal; MV: Mesiobuccal.

**Table 4 jcm-12-01383-t004:** Linear measurements corresponding to upper right molars in the presence and absence of transverse bone discrepancy.

			Discrepancy		*p*-Value
		Total	With Transverse Discrepancy	Without Transverse Discrepancy	
16-OSSEOUS V (DV)	N	120	60	60	
	Mean	2.55	2.59	2.52	0.627
	SD	0.79	0.82	0.75	
	95%CI	(2.41, 2.69)	(2.37, 2.8)	(2.32, 2.71)	
16-GINGIVAL(DV)	Mean	3.29	3.32	3.25	0.714
	SD	0.9	0.98	0.81	
	95%CI	(3.12, 3.45)	(3.06, 3.57)	(3.04, 3.47)	
16-OSSEOUS V (MV)	Mean	1.48	1.51	1.46	0.631
	SD	0.63	0.64	0.62	
	95%CI	(1.37, 1.6)	(1.35, 1.68)	(1.3, 1.62)	
16-OSSEOUS P	Mean	1.12	1.1	1.14	0.619
	SD	0.43	0.4	0.46	
	95%CI	(1.04, 1.2)	(1, 1.2)	(1.02, 1.26)	
16-GINGIVAL P	Mean	4.05	4.21	3.9	0.095
	SD	1	1.01	0.98	
	95%CI	(3.87, 4.24)	(3.95, 4.47)	(3.65, 4.15)	
17-OSSEOUS V (DV)	Mean	2.66	2.66	2.66	0.998
	SD	1.13	1.21	1.07	
	95%CI	(2.46, 2.87)	(2.35, 2.97)	(2.39, 2.94)	
17-OSSEOUS P	Mean	1.69	1.73	1.65	0.621
	SD	0.93	0.9	0.96	
	95%CI	(1.52, 1.86)	(1.5, 1.97)	(1.4, 1.9)	
17-OSSEOUS (MV)	Mean	2.45	2.47	2.43	0.827 (ANOVA)
	SD	1.01	1.03	1.01	
	95%CI	(2.27, 2.64)	(2.21, 2.74)	(2.17, 2.69)	

Estimated marginal means, SD and 95%CI, significance between subjects effects test (MANOVA); significant *p* ≤ 0.05, significant *p* ≤ 0.01. V: Buccal; P: Palatine; DV: Distobuccal; MV: Mesiobuccal.

**Table 5 jcm-12-01383-t005:** Linear measurements corresponding to upper left molars in the presence and absence of transverse bone discrepancy.

			Discrepancy		*p*-Value
		Total	With Transverse Discrepancy	Without Transverse Discrepancy	
26-OSSEOUS V (DV)	Mean	2.53	2.54	2.52	0.911
	SD	0.83	0.82	0.85	
	95%CI	(2.38, 2.68)	(2.32, 2.75)	(2.3, 2.74)	
26-GINGIVAL V (DV)	Mean	3.32	3.48	3.16	0.116
	SD	1.09	1.01	1.15	
	95%CI	(3.12, 3.52)	(3.22, 3.74)	(2.87, 3.46)	
26-OSSEOUS V (MV)	Mean	1.95	2.14	1.75	0.059
	SD	1.12	1.2	1.01	
	95%CI	(1.74, 2.15)	(1.83, 2.45)	(1.49, 2.01)	
26-OSSEOUS P	Mean	2.17	2.62	1.72	0.002 **
	SD	1.6	1.74	1.31	
	95%CI	(1.88, 2.46)	(2.17, 3.07)	(1.38, 2.06)	
26-GINGIVAL P	Mean	4.37	4.99	3.75	0.000 **
	SD	1.53	1.39	1.42	
	95%CI	(4.09, 4.65)	(4.63, 5.35)	(3.38, 4.12)	
27-OSSEOUS (DV)	Mean	2.59	2.77	2.41	0.064
	SD	1.08	1.21	0.9	
	95%CI	(2.39, 2.78)	(2.46, 3.08)	(2.17, 2.64)	
27-OSSEOUS (MV)	Mean	2.29	2.34	2.24	0.576
	SD	0.92	0.92	0.93	
	95%CI	(2.12, 2.45)	(2.1, 2.57)	(2, 2.48)	
27-OSSEOUS P	Mean	1.61	1.53	1.7	0.276
	SD	0.82	0.73	0.89	
	95%CI	(1.47, 1.76)	(1.34, 1.72)	(1.46, 1.93)	

Estimated marginal means, SD and 95%CI, significance between subjects effects test (MANOVA); significant *p* ≤ 0.05, ** significant *p* ≤ 0.01. V: Buccal; P: Palatine; DV: Distobuccal; MV: Mesiobuccal.

**Table 6 jcm-12-01383-t006:** Linear measurements corresponding to upper right molars in the different vertical facial patterns, in the presence and absence of transverse bone discrepancy.

		Pattern	Brachyfacial		Dolichofacial		Mesofacial		*p*-Value
		Total	With Transverse Discrepancy	Without Transverse Discrepancy	With Transverse Discrepancy	Without Transverse Discrepancy	With Transverse Discrepancy	Without Transverse Discrepancy	
16-OSSEOUS V (DV)	N	120	20	20	20	20	20	20	
	Mean	2.55	2.5	2.53	2.48	2.44	2.78	2.58	0.815
	SD	0.79	0.68	0.9	1	0.84	0.75	0.48	
	95%CI	(2.41, 2.69)	(2.18, 2.82)	(2.1, 2.95)	(2.02, 2.95)	(2.05, 2.83)	(2.42, 3.13)	(2.36, 2.81)	
16-GINGIVAL(DV)	Mean	3.29	3.24	3.3	3.18	3.15	3.53	3.32	0.8
	SD	0.9	0.89	1.01	1.11	0.86	0.95	0.52	
	95%CI	(3.12, 3.45)	(2.82, 3.66)	(2.82, 3.77)	(2.66, 3.7)	(2.74, 3.55)	(3.08, 3.98)	(3.08, 3.56)	
16-OSSEOUS V (MV)	Mean	1.48	1.2	1.34	1.03	1.09	1.07	0.99	0.637
	SD	0.63	0.47	0.55	0.3	0.41	0.39	0.34	
	95%CI	(1.37, 1.6)	(0.98, 1.42)	(1.08, 1.59)	(0.89, 1.17)	(0.9, 1.28)	(0.88, 1.25)	(0.83, 1.15)	
16-OSSEOUS P	Mean	1.12	3.94	3.69	4.71	3.96	3.97	4.06	0.513
	SD	0.43	0.9	1.02	0.96	0.88	1.01	1.05	
	95%CI	(1.04, 1.2)	(3.52, 4.36)	(3.21, 4.17)	(4.27, 5.16)	(3.55, 4.37)	(3.49, 4.44)	(3.57, 4.55)	
16-GINGIVAL P	Mean	4.05	1.66	1.47	1.32	1.4	1.56	1.5	0.15
	SD	1	0.55	0.71	0.65	0.7	0.71	0.46	
	95%CI	(3.87, 4.24)	(1.41, 1.92)	(1.14, 1.8)	(1.02, 1.62)	(1.07, 1.72)	(1.22, 1.89)	(1.29, 1.72)	
17-OSSEOUS V (DV)	Mean	2.66	2.61	2.42	2.64	2.92	2.74	2.64	0.621
	SD	1.13	1.02	1.18	1.28	0.98	1.35	1.03	
	95%CI	(2.46, 2.87)	(2.13, 3.09)	(1.87, 2.98)	(2.04, 3.24)	(2.46, 3.38)	(2.11, 3.37)	(2.16, 3.12)	
17-OSSEOUS P	Mean	1.69	1.83	1.66	1.65	1.94	1.72	1.35	0.279
	SD	0.93	0.91	0.95	0.92	1.18	0.9	0.61	
	95%CI	(1.52, 1.86)	(1.41, 2.26)	(1.22, 2.11)	(1.22, 2.09)	(1.39, 2.49)	(1.29, 2.14)	(1.07, 1.63)	
17-OSSEOUS (MV)	Mean	2.45	2.35	2.53	2.34	2.43	2.74	2.34	0.389
	SD	1.01	0.95	1.15	1	0.82	1.12	1.06	
	95%CI	(2.27, 2.64)	(1.9, 2.79)	(1.99, 3.07)	(1.87, 2.81)	(2.04, 2.82)	(2.21, 3.26)	(1.84, 2.83)	

Estimated marginal means, SD and 95%CI, significance between subjects effects test (MANOVA); significant *p* ≤ 0.05, significant *p* ≤ 0.01. V: Buccal; P: Palatine; DV: Distobuccal; MV: Mesiobuccal.

**Table 7 jcm-12-01383-t007:** Linear measurements corresponding to upper left molars in the different vertical facial patterns, in the presence and absence of transverse bone discrepancy.

		Pattern	Brachyfacial		Dolichofacial		Mesofacial		*p*-Value
		Total	With Transverse Discrepancy	Without Transverse Discrepancy	With Transverse Discrepancy	Without Transverse Discrepancy	With Transverse Discrepancy	Without Transverse Discrepancy	
26-OSSEOUS V (DV)	Mean	2.53	2.61	2.37	2.47	2.73	2.53	2.46	0.417
	SD	0.83	0.85	0.94	0.91	0.95	0.72	0.61	
	95%CI	(2.38, 2.68)	(2.21, 3.01)	(1.93, 2.81)	(2.04, 2.9)	(2.28, 3.17)	(2.19, 2.86)	(2.17, 2.74)	
26-GINGIVAL V (DV)	Mean	3.32	3.58	3.23	3.4	3.1	3.45	3.16	0.990
	SD	1.09	1.15	1.03	1.07	1.21	0.83	1.26	
	95%CI	(3.12, 3.52)	(3.04, 4.12)	(2.74, 3.71)	(2.9, 3.9)	(2.54, 3.67)	(3.07, 3.84)	(2.57, 3.75)	
26-OSSEOUS V (MV)	Mean	1.95	2.46	1.6	1.82	2.12	2.13	1.53	0.048 *
	SD	1.12	1.15	0.94	1.25	1.25	1.17	0.69	
	95%CI	(1.74, 2.15)	(1.92, 3)	(1.16, 2.04)	(1.24, 2.4)	(1.54, 2.71)	(1.59, 2.68)	(1.2, 1.85)	
26-OSSEOUS P	Mean	2.17	2.67	1.29	2.21	2.44	2.99	1.43	0.016
	SD	1.6	1.48	0.48	1.97	1.85	1.74	0.92	
	95%CI	(1.88, 2.46)	(1.98, 3.36)	(1.07, 1.52)	(1.29, 3.14)	(1.58, 3.31)	(2.17, 3.8)	(1, 1.86)	
26-GINGIVAL P	Mean	4.37	4.88	3.75	5.19	3.79	4.9	3.71	0.909
	SD	1.53	1.33	1.22	1.36	1.59	1.51	1.51	
	95%CI	(4.09, 4.65)	(4.26, 5.51)	(3.18, 4.33)	(4.55, 5.82)	(3.05, 4.54)	(4.19, 5.61)	(3, 4.41)	
27-OSSEOUS (DV)	Mean	2.59	2.65	2.21	2.97	2.55	2.69	2.45	0.890
	SD	1.08	0.88	0.94	1.13	0.84	1.57	0.93	
	95%CI	(2.39, 2.78)	(2.24, 3.06)	(1.77, 2.65)	(2.45, 3.5)	(2.16, 2.94)	(1.95, 3.42)	(2.02, 2.89)	
27-OSSEOUS (MV)	Mean	2.29	2.38	2.21	2.25	2.43	2.37	2.08	0.500
	SD	0.92	0.99	1.11	0.61	0.94	1.13	0.7	
	95%CI	(2.12, 2.45)	(1.92, 2.84)	(1.69, 2.73)	(1.96, 2.54)	(1.99, 2.87)	(1.85, 2.9)	(1.75, 2.41)	
27-OSSEOUS P	Mean	1.61	1.6	1.66	1.39	1.69	1.61	1.73	0.792
	SD	0.82	0.64	0.77	0.74	0.84	0.82	1.08	
	95%CI	(1.47, 1.76)	(1.3, 1.9)	(1.3, 2.02)	(1.04, 1.73)	(1.3, 2.08)	(1.22, 1.99)	(1.23, 2.24)	
	95%CI	(2.27, 2.64)	(1.9, 2.79)	(1.99, 3.07)	(1.87, 2.81)	(2.04, 2.82)	(2.21, 3.26)	(1.84, 2.83)	

Estimated marginal means, SD and 95%CI, significance between subjects effects test (MANOVA); * significant *p* ≤ 0.05, significant *p* ≤ 0.01. V: Buccal; P: Palatine; DV: Distobuccal; MV: Mesiobuccal.

## Data Availability

The data described in this study are available upon request to the corresponding author. The data are not publicly available, due to the need to observe patient confidentiality.

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
