# Peer review of "Soft- and Hard-Tissue Thicknesses in Patients with Different Vertical Facial Patterns and the Transverse Deficiencies, An Integrated CBCT-3D Digital Model Analysis"

_jcm, 2023, doi:10.3390/jcm12041383_

Round 1
Reviewer 1 Report
Thanks very much for inviting me to review this paper. The content is interesting. However, several issues require addressing before the paper can be further considered for publication.
Title
1- The title does not reflect the content of this paper. The authors should have stated that they were studying widths (transverse distances or soft- and hard-tissue thicknesses) instead of using the vague word 'distances'. Also, the authors should have mentioned both factors together without indicating that there was a correlation analysis between the facial pattern and the transverse discrepancy. In other words, the title should mention both factors together (i.e., the vertical facial pattern and the transverse deficiency). My third point is the use of the word "superimposition". The authors should use the term "Combined CBCT-3D digital model analysis", "Merged CBCT and 3D-digital-model analysis", or "integrated CBCT-3D digital model analysis". The study design is preferred to be mentioned at the end of the title after using a colon."
Abstract
2- Line 25: please try to use another word than "superimposition". Please make this change here and everywhere in the manuscript.
Introduction
3- Is the WALA ridge a limit of the transverse expansion of the dentoalveolar bone? What is the evidence beyond this assumption?
4- Please justify the onset of this research project by showing the shortcomings of previous studies in this regard.
Materials and Methods
5- Line 75: what are the reasons for having this database of CBCT images?
6- Line 79: how did you exclude patients with posterior crossbites? As you know, many patients are expected to have a maxillo-mandibular transverse deficiency and, at the same time, posterior crossbite. Excluding patients with posterior crossbites may lead to insufficient patients in the second subgroup (patients with a transverse deficiency). The normal scenario is to include those patients with posterior crossbites.
7- Line 88: please give the details of your assumptions using G*Power version 3.1.
8- Line 90: what is your sampling frame (i.e., the number of CBCT images in your database deemed suitable for your study)? What kind of sampling did you use to build up your six groups?
9- Line 108: information about the product is required (product name, company name, city, country).
10- Line 109: information about the product is required (product name, company name, city, country).
11- Line 112: a FOV of 8 x 8 cm is insufficient for your study. The images presented show that you were using a bigger FOV. Please clarify this point.
12- Line 126: please insert information about the product (product name, company name, city, country).
13- Line 137: what do you mean by the WALA-WALA reference line?
14- Line 164: please insert information about the product (product name, company name, city, country).
Results
15- Please explain all the abbreviations used in the tables in their footnotes.
Discussion
16- Line 276: Please augment your arguments with the data from this source: Alhawasli RY, Ajaj MA, Hajeer MY, Al-Zahabi AMR, Mahaini L. Volumetric Analysis of the Jaws in Skeletal Class I and III Patients with Different Facial Divergence Using CBCT Imaging. Radiol Res Pract. 2022 May 28;2022:2416555. doi: 10.1155/2022/2416555. PMID: 35668737; PMCID: PMC9167144.
17- Line 287: Please try to use the results found in this study: El-Schallah AA, Ajaj MA, Hajeer MY. The Relationship Between Vertical Facial Type and Maxillary Anterior Alveolar Angle in Adults Using Cone-Beam Computed Tomography. Cureus. 2022 Oct 16;14(10):e30356. doi: 10.7759/cureus.30356. PMID: 36258803; PMCID: PMC9573689
18- Line 290: The answer to the raised question in this sentence is found in the paper mentioned above (El-Schallah et al., 2022).
19- Line 334: The dependence on WALA does not always represent the mandibular basal bone. The authors are asked to read this paper that gives another basal bone definition using CBCT data and to try to incorporate its findings with the current discussion: Al-Hilal LH, Sultan K, Hajeer MY, Mahmoud G, Wanli AA. An Evaluation of Mandibular Dental and Basal Arch Dimensions in Class I and Class II Division 1 Adult Syrian Patients using Cone-beam Computed Tomography. J Contemp Dent Pract. 2018 Apr 1;19(4):431-437. PMID: 29728549.
20- Line 335: The authors should also mention that volumetric assessment of the available bone between the boundaries of the alveolar bone and root surfaces could be another indicator. Volumetric assessment is becoming an important tool in recent research in orthodontics in patients with different vertical, sagittal, or transverse relationships (A reference could be cited here: Nahas BD, Hajeer MY, Ajaj MA, Alsabbagh AY. Volumetric Analysis of the Jaws in Skeletal Class I and Class II Patients using CBCT and Derived Lateral Cephalograms. Journal of Clinical & Diagnostic Research. 2017 Dec 1;11(12).
Conclusions
Fine.
Author Response
Dear reviewer,
Thank you very much for your comments referred to our manuscript, which have been addressed on a point-by-point basis below, and have been incorporated to the revised version of the article:
Comments and Suggestions for Authors
Title
1-The title does not reflect the content of this paper. The authors should have stated that they were studying widths (transverse distances or soft- and hard-tissue thicknesses) instead of using the vague word 'distances'. Also, the authors should have mentioned both factors together without indicating that there was a correlation analysis between the facial pattern and the transverse discrepancy. In other words, the title should mention both factors together (i.e., the vertical facial pattern and the transverse deficiency). My third point is the use of the word "superimposition". The authors should use the term "Combined CBCT-3D digital model analysis", "Merged CBCT and 3D-digital-model analysis", or "integrated CBCT-3D digital model analysis". The study design is preferred to be mentioned at the end of the title after using a colon."
Reply: The title has been reworded: “Soft- and hard-tissue thicknesses in patients with different vertical facial patterns and the transverse deficiencies. An integrated CBCT-3D digital model analysis”
Abstract
2-Line 25: please try to use another word than "superimposition". Please make this change here and everywhere in the manuscript.
Reply: We have replaced “superimposition” with “integration” in the Abstract and throughout the manuscript.
Introduction
3-Is the WALA ridge a limit of the transverse expansion of the dentoalveolar bone? What is the evidence beyond this assumption?
Reply: The WALA ridge in the mandible could be used as an apical bone reference and could be employed for the individualized arch shape of the patient, thus determining the transversal dentoalveolar modification needs. The paragraph has been changed and additional literature references have been added. “The WALA ridge can be used to determine the apical base of the mandibular bone; it is useful for designing an individualized dental arch shape, and can be used in the maxillary arch as a reference for maxillary width [12-14]”. Line 61.
Ronay V, Miner RM, Will LA, Arai K. Mandibular arch form: The relationship between dental and basal anatomy. Am. J. Orthod. Dentofac. Orthop. 2008;134(3):430–8.
Ball RL, Miner RM, Will LA, Arai K. Comparison of dental and apical base arch forms in Class II Division 1 and Class i malocclusions. Am. J. Orthod. Dentofac. Orthop. 2010;138(1):41–50. Available at: http://dx.doi.org/10.1016/j.ajodo.2008.11.026.
We have also added the following: “Determination of the basal bone level remains a confusing issue among authors, as there is no single criterion. Andrews [15] proposed the WALA points to estimate the width of the basal arch, but this can be altered by soft tissue modifications. Al-Hilal et al. [20] in turn proposed a novel method for analyzing the mandibular basal width at the level of the canines and first molars that would not be affected by the soft tissues. They chose the junction of the middle third with the apical third at canine level as apical reference, and for the transverse measurements selected the midpoint between the buccal and lingual cortical layers to perform the basal measures in the axial view. Line: 287
Al-Hilal LH, Sultan K, Hajeer MY, Mahmoud G, Wanli AA. An Evaluation of Mandibular Dental and Basal Arch Dimensions in Class I and Class II Division 1 Adult Syrian Patients using Cone-beam Computed Tomography. J Contemp Dent Pract. 2018 Apr 1;19(4):431-437
4- Please justify the onset of this research project by showing the shortcomings of previous studies in this regard.
Reply: The following argument has been added: “Although a number of studies have examined the association between cortical bone thickness and vertical facial pattern [6,8,16,17], no analyses have been made relating it to the transverse bone dimensions. Cone-beam computed tomography based studies likewise have not analyzed gingival thickness in different vertical facial patterns with and without transverse deficiency.” Line: 66
“Thus, the present study was carried out to compare the distances to the WALA ridge and WALA bone from the buccal root surface, and from the palatine root surface of upper molars to the cortical bone and buccal and palatine gingival portion, in the different facial patterns, in the presence and absence of transverse bone discrepancies, and the combination facial pattern - transverse bone discrepancy.” Line: 71
Materials and Methods
5- Line 75: what are the reasons for having this database of CBCT images?
Reply: The selected patients belong to the database of the Master of Orthodontics of the Universidad Católica de Murcia (Spain), where they are seen for orthodontic treatment. Those patients subjected to CBCT due to requirements of their treatment were considered candidates for inclusion in the study.
6- Line 79: how did you exclude patients with posterior crossbites? As you know, many patients are expected to have a maxillo-mandibular transverse deficiency and, at the same time, posterior crossbite. Excluding patients with posterior crossbites may lead to insufficient patients in the second subgroup (patients with a transverse deficiency). The normal scenario is to include those patients with posterior crossbites.
Reply: Patients with posterior crossbite were discarded because we aimed to determine whether, in the presence of transverse bone but not dental discrepancy, any of the facial patterns presented lesser or greater bone and/or gingival thicknesses that could affect clinical decision in planning treatment. The results of this study are of clinical relevance, since they suggest that patients with transverse bone discrepancies and a mesofacial or brachyfacial pattern will have a better prognosis than dolichofacial individuals, as the latter would have a more limited dental compensation capacity. This could imply an increased risk of bone dehiscence in dolichofacial patients in the event dentoalveolar expansion is planned to solve a transverse bone discrepancy in an individual without posterior cross-bite [28].” Line: 345
7- Line 88: please give the details of your assumptions using G*Power version 3.1.
Reply: The analyzed MANOVA model comprises two independent variables (problem and pattern) that define 6 groups, and there are three dependent variables in each model, selecting a medium effect size (0.1) (Cohen suggested different benchmarks for MANOVA, with a medium-sized effect having an f-squared of 0.15. See Steyn, H. S., Jr. & Ellis, S. M. (2009). Estimating an effect size in one-way multivariate analysis of variance (MANOVA). Multivariate Behavioral Research, 44(1), 106-129.)
8- Line 90: what is your sampling frame (i.e., the number of CBCT images in your database deemed suitable for your study)? What kind of sampling did you use to build up your six groups?
Reply: We have added more information about the sampling used: “The included cases belonged to the patient database of the Master of Orthodontics of the Universidad Católica de Murcia (Spain). The database contained a total of 286 histories of patients seen in the Department of Orthodontics during the period between March 2018 and January 2022. Following an analysis of the necessary records and of compliance with the inclusion / exclusion criteria, a sample of 178 potentially recruitable patients was established, in which an analysis was made of transverse discrepancy in the CBCT study and of the facial pattern from lateral teleradiography of the skull, obtaining: 57 brachyfacial patterns (32 with transverse discrepancy and 25 without discrepancy) 66 mesofacial patterns (39 with transverse discrepancy and 27 without discrepancy) and 55 dolichofacial patterns (29 with transverse discrepancy and 26 without discrepancy). Random selection was then made (based on a computer-generated list) to establish the following 6 groups: (I) brachyfacial with transverse bone discrepancy (n=20; 9 males and 11 females), (II) brachyfacial without transverse bone discrepancy (n=20; 8 males and 12 females), (III) mesofacial with transverse bone discrepancy (n=20; 8 males and 12 females), (IV) mesofacial without transverse bone discrepancy (n=20; 9 males and 11 females), (V) dolichofacial with transverse bone discrepancy (n=20; 7 males and 13 females), and (VI) dolichofacial without transverse bone discrepancy (n=20; 4 males and 16 females). The MANOVA models were adjusted a posteriori for sex and age.” Line: 93.
9- Line 108: information about the product is required (product name, company name, city, country).
Reply: Further details have been added on the product “CS 3600 scanner (Carestream Dental®, Atlanta, USA)”. Line: 124
10- Line 109: information about the product is required (product name, company name, city, country).
Reply: Further details have been added on the product “Orthophos SL 2D / 3D (Dentsply Sirona®, Charlotte, USA).” Line: 125
11- Line 112: a FOV of 8 x 8 cm is insufficient for your study. The images presented show that you were using a bigger FOV. Please clarify this point.
Reply: The FOV size has been corrected: “The volume area of the object/field of view (FoV) corresponded to 11 cm x 10 cm”. Line: 128.
12- Line 126: please insert information about the product (product name, company name, city, country).
Reply: Further details have been added on the product “NemoStudio 2018 (Nemotec®, Madrid, Spain)” Line: 129.
13- Line 137: what do you mean by the WALA-WALA reference line?
Reply: A horizontal reference line was used on which to perform the linear measurements, affording precision, and using it to validate the method at the time of intra-operator calibration. It is a horizontal line joining the right WALA point of the patient with the left WALA point.
14- Line 164: please insert information about the product (product name, company name, city, country).
Reply: Further details have been added on the product “SPSS version 25 statistical package for MS Windows (IBM Corp., Armonk, New York, USA)”. Line: 180.
Results
15- Please explain all the abbreviations used in the tables in their footnotes.
Reply: The abbreviations have been explained in the tables. V: Buccal; P: Palatine; DV: Distobuccal; MV: Mesiobuccal.
Discussion
16- Line 276: Please augment your arguments with the data from this source: Alhawasli RY, Ajaj MA, Hajeer MY, Al-Zahabi AMR, Mahaini L. Volumetric Analysis of the Jaws in Skeletal Class I and III Patients with Different Facial Divergence Using CBCT Imaging. Radiol Res Pract. 2022 May 28;2022:2416555. doi: 10.1155/2022/2416555. PMID: 35668737; PMCID: PMC9167144.
Reply: The following has been added with the corresponding reference: “Alhawasli et al. [22], in their volumetric study of the maxilla and mandible, also found mandibular bone volume in hypodivergent class III individuals to be significantly greater than in hyperdivergent class III patients.” Line: 303.
Alhawasli RY, Ajaj MA, Hajeer MY, Al-Zahabi AMR, Mahaini L. Volumetric Analysis of the Jaws in Skeletal Class I and III Patients with Different Facial Divergence Using CBCT Imaging. Radiol Res Pract. 2022 May 28;2022:2416555.
17- Line 287: Please try to use the results found in this study: El-Schallah AA, Ajaj MA, Hajeer MY. The Relationship Between Vertical Facial Type and Maxillary Anterior Alveolar Angle in Adults Using Cone-Beam Computed Tomography. Cureus. 2022 Oct 16;14(10):e30356. doi: 10.7759/cureus.30356. PMID: 36258803; PMCID: PMC9573689
Reply: The following has been added with the corresponding reference: “These different bone thicknesses of the anterior maxillary region in the different vertical facial patterns may also be related to differences in inclination of the anterior maxillary alveolar process, since it has been seen that brachyfacial patterns are characterized by a more buccal inclination than in dolichofacial individuals [23]”. Line 317.
El-Schallah AA, Ajaj MA, Hajeer MY. The Relationship Between Vertical Facial Type and Maxillary Anterior Alveolar Angle in Adults Using Cone-Beam Computed Tomography. Cureus. 2022 Oct 16;14(10)
18- Line 290: The answer to the raised question in this sentence is found in the paper mentioned above (El-Schallah et al., 2022).
Reply: It has been specified that reference is made to the buccolingual inclination of the molars (bucco-lingual molar axis), not of the alveolar process: “The literature does not clarify whether the different vertical facial patterns are characterized by differences in buccolingual inclination of the upper molars”. Line: 329.
19- Line 334: The dependence on WALA does not always represent the mandibular basal bone. The authors are asked to read this paper that gives another basal bone definition using CBCT data and to try to incorporate its findings with the current discussion: Al-Hilal LH, Sultan K, Hajeer MY, Mahmoud G, Wanli AA. An Evaluation of Mandibular Dental and Basal Arch Dimensions in Class I and Class II Division 1 Adult Syrian Patients using Cone-beam Computed Tomography. J Contemp Dent Pract. 2018 Apr 1;19(4):431-437. PMID: 29728549.
Reply: The following has been added with the corresponding reference: “Determination of the basal bone level remains a confusing issue among authors, as there is no single criterion. Andrews [15] proposed the WALA points to estimate the width of the basal arch, but this can be altered by soft tissue modifications. Al-Hilal et al. [20] in turn proposed a novel method for analyzing the mandibular basal width at the level of the canines and first molars that would not be affected by the soft tissues. They chose the junction of the middle third with the apical third at canine level as apical reference, and for the transverse measurements selected the midpoint between the buccal and lingual cortical layers to perform the basal measures in the axial view. On comparing the mandibular basal width between males and females with class I and class II division 1 malocclusions, they recorded greater basal widths in males versus females, and the basal widths at canine level were moreover smaller in subjects with class II division 1 malocclusions than in those with class I malocclusions.” Line: 287.
Al-Hilal LH, Sultan K, Hajeer MY, Mahmoud G, Wanli AA. An Evaluation of Mandibular Dental and Basal Arch Dimensions in Class I and Class II Division 1 Adult Syrian Patients using Cone-beam Computed Tomography. J Contemp Dent Pract. 2018 Apr 1;19(4):431-437
20- Line 335: The authors should also mention that volumetric assessment of the available bone between the boundaries of the alveolar bone and root surfaces could be another indicator. Volumetric assessment is becoming an important tool in recent research in orthodontics in patients with different vertical, sagittal, or transverse relationships (A reference could be cited here: Nahas BD, Hajeer MY, Ajaj MA, Alsabbagh AY. Volumetric Analysis of the Jaws in Skeletal Class I and Class II Patients using CBCT and Derived Lateral Cephalograms. Journal of Clinical & Diagnostic Research. 2017 Dec 1;11(12).
Reply: The following has been added with the corresponding reference:: Nahas et al. [21], in their volumetric study of the maxilla and mandible, considered elimination of the maxillary and mandibular crowns for performing the bone volumetric measurements. Line: 297.
Nahas BD, Hajeer MY, Ajaj MA, Alsabbagh AY. Volumetric Analysis of the Jaws in Skeletal Class I and Class II Patients using CBCT and Derived Lateral Cephalograms. Journal of Clinical & Diagnostic Research. 2017 Dec 1;11(12).
Conclusions
Fine.

Reviewer 2 Report
I found this research manuscript interesting and relevant. It is well-written and comprehensible. I have few minor comments
1. Please give details about measures taken to prevent bias. The patients included in the study were selected after reviewing their radiographic records. How blinding was performed. Was randomization done?
2. Line 92-93: What do these numbers in the bracket signify (n=40; 51 males and 69 females), dolichofacial (n=40; 33 males and 87 females) or brachyfacial (n=40; 51 males and 69 females).
3. Table 1: Will the High Percentage of Females in Dolichofacial group (72%) as compared to Mesofacial group (57.5%) have any effect on the study outcome? As the study is a retrospective study, is it possible to keep the percentage of selected female and male patients almost the same for all the groups by searching a larger database.
Even the median age of patients’ selected in the dolichofacial group is lower as compared to other groups. Can it affect the outcome of the study?
4. Please add clinical significance of the study.
Author Response
Dear reviewer,
Thank you very much for your comments referred to our manuscript, which have been addressed on a point-by-point basis below, and have been incorporated to the revised version of the article.
I found this research manuscript interesting and relevant. It is well-written and comprehensible. I have few minor comments
1.Please give details about measures taken to prevent bias. The patients included in the study were selected after reviewing their radiographic records. How blinding was performed. Was randomization done?
Reply: We have added more information about the sampling used: “The included cases belonged to the patient database of the Master of Orthodontics of the Universidad Católica de Murcia (Spain). The database contained a total of 286 histories of patients seen in the Department of Orthodontics during the period between March 2018 and January 2022. Following an analysis of the necessary records and of compliance with the inclusion / exclusion criteria, a sample of 178 potentially recruitable patients was established, in which an analysis was made of transverse discrepancy in the CBCT study and of the facial pattern from lateral teleradiography of the skull, obtaining: 57 brachyfacial patterns (32 with transverse discrepancy and 25 without discrepancy) 66 mesofacial patterns (39 with transverse discrepancy and 27 without discrepancy) and 55 dolichofacial patterns (29 with transverse discrepancy and 26 without discrepancy). Random selection was then made (based on a computer-generated list) to establish the following 6 groups: (I) brachyfacial with transverse bone discrepancy (n=20; 9 males and 11 females), (II) brachyfacial without transverse bone discrepancy (n=20; 8 males and 12 females), (III) mesofacial with transverse bone discrepancy (n=20; 8 males and 12 females), (IV) mesofacial without transverse bone discrepancy (n=20; 9 males and 11 females), (V) dolichofacial with transverse bone discrepancy (n=20; 7 males and 13 females), and (VI) dolichofacial without transverse bone discrepancy (n=20; 4 males and 16 females). The MANOVA models were adjusted a posteriori for sex and age.” Line: 93.
- Line 92-93: What do these numbers in the bracket signify (n=40; 51 males and 69 females), dolichofacial (n=40; 33 males and 87 females) or brachyfacial (n=40; 51 males and 69 females).
Reply: The error has been corrected: mesofacial (n=40; 17 males and 23 females), dolichofacial (n=40; 11 males and 29 females) or brachyfacial (n=40; 17 males and 23 females)”. These are data obtained from Table 1.
- Table 1: Will the High Percentage of Females in Dolichofacial group (72%) as compared to Mesofacial group (57.5%) have any effect on the study outcome? As the study is a retrospective study, is it possible to keep the percentage of selected female and male patients almost the same for all the groups by searching a larger database.
Even the median age of patients’ selected in the dolichofacial group is lower as compared to other groups. Can it affect the outcome of the study?
Reply: There may have been minor effects upon the results in this subgroup, due to the large percentage of females, as we were unable a priori to homogenize for sex and age, though a posteriori the MANOVA models were adjusted for age and sex. Likewise, although the selected patients were over 18 years of age and were regarded as adults, there may have been minor effects that were also subsequently adjusted in the MANOVA. The fact of making no distinction according to age and sex has been included as a limitation of our study. Line: 373.
- Please add clinical significance of the study.
Reply: The clinical importance of the study has been pointed out: “The results of this study are of clinical relevance, since they suggest that patients with transverse bone discrepancies and a mesofacial or brachyfacial pattern will have a better prognosis than dolichofacial individuals, as the latter would have a more limited dental compensation capacity. This could imply an increased risk of bone dehiscence in dolichofacial patients in the event dentoalveolar expansion is planned to solve a transverse bone discrepancy in an individual without posterior cross-bite [28].” Line: 345.
Renkema AM, Fudalej PS, Renkema A, Kiekens R, Katsaros C. Development of labial gingival recessions in orthodontically treated patients. Am J Orthod Dentofac Orthop 2013;143(2):206–12.

Reviewer 3 Report
- An approval of an Ethics Committee is necessary and informed consent signed by the patients.
- How was the sample size of 120 individuals obtained and 40 individuals in each of the three groups?
- The study shows a novel method to evaluate the distances from the upper first and second molars to the WALA ridge. Therefore, a comparison to the previous method should provide conclusions to accept it as a better way to have clinical applications. The conclusions of this study give data on the new method, but we do not know if these give us a better and more exact way to obtain a proper diagnose.
Author Response
Dear reviewer,
Thank you very much for your comments referred to our manuscript, which have been addressed on a point-by-point basis below, and have been incorporated to the revised version of the article:
Comments and Suggestions for Authors
1.An approval of an Ethics Committee is necessary and informed consent signed by the patients.
Reply: The present study was approved by the Ethics Committee of the Universidad Católica de Murcia, with internal code CE012006; date 31/01/2020, and patient informed consent was obtained. Line: 392
2.How was the sample size of 120 individuals obtained and 40 individuals in each of the three groups?
Reply: We have added more information about the sampling used: “The included cases belonged to the patient database of the Master of Orthodontics of the Universidad Católica de Murcia (Spain). The database contained a total of 286 histories of patients seen in the Department of Orthodontics during the period between March 2018 and January 2022. Following an analysis of the necessary records and of compliance with the inclusion / exclusion criteria, a sample of 178 potentially recruitable patients was established, in which an analysis was made of transverse discrepancy in the CBCT study and of the facial pattern from lateral teleradiography of the skull, obtaining: 57 brachyfacial patterns (32 with transverse discrepancy and 25 without discrepancy) 66 mesofacial patterns (39 with transverse discrepancy and 27 without discrepancy) and 55 dolichofacial patterns (29 with transverse discrepancy and 26 without discrepancy). Random selection was then made (based on a computer-generated list) to establish the following 6 groups: (I) brachyfacial with transverse bone discrepancy (n=20; 9 males and 11 females), (II) brachyfacial without transverse bone discrepancy (n=20; 8 males and 12 females), (III) mesofacial with transverse bone discrepancy (n=20; 8 males and 12 females), (IV) mesofacial without transverse bone discrepancy (n=20; 9 males and 11 females), (V) dolichofacial with transverse bone discrepancy (n=20; 7 males and 13 females), and (VI) dolichofacial without transverse bone discrepancy (n=20; 4 males and 16 females). The MANOVA models were adjusted a posteriori for sex and age. The distribution of the study sample is reflected in Table 1.” Line: 93.
3.The study shows a novel method to evaluate the distances from the upper first and second molars to the WALA ridge. Therefore, a comparison to the previous method should provide conclusions to accept it as a better way to have clinical applications. The conclusions of this study give data on the new method, but we do not know if these give us a better and more exact way to obtain a proper diagnose.
Reply: In effect, in this study we cannot conclude that it would be a more exact method than analyzing only the CBCT data, though it is a method allowing joint analysis of the bone and gingival thicknesses in a precise manner, and can be used in those cases characterized by CBCT images with distortions or poor quality impeding correct visualization of the gingival structures. This study also carries out measurements from the root structure to the WALA ridge, something not done in previous studies, based on integration of the CBCT images (DICOM files) and the digital model images (STL files). Nevertheless, it would be interesting for future research to conduct comparative studies between CBCT analysis alone and analysis based on CBCT-STL integration.

Round 2
Reviewer 1 Report
Thanks to the authors for successfully addressing my raised points in my first manuscript review.
Author Response
Thank you very much for your comments that have served to improve my manuscript.
Reviewer 3 Report
Two of the requeriments have not been attended
- An approval of an Ethics Committee is necessary and informed consent signed by the patients.
- The study shows a novel method to evaluate the distances from the upper first and second molars to the WALA ridge. Therefore, a comparison to the previous method should provide conclusions to accept it as a better way to have clinical applications. The conclusions of this study give data on the new method, but we do not know if these give us a better and more exact way to obtain a proper diagnose.
Author Response
Dear reviewer,
Thank you very much for your comments referred to our manuscript, which have been addressed on a point-by-point basis below:
Comments and Suggestions for Authors
Two of the requeriments have not been attended
- An approval of an Ethics Committee is necessary and informed consent signed by the patients.
Reply: The following information is in the manuscript: "Institutional Review Board Statement: The present study was approved by the Ethics Committee of the Catholic University of Murcia, with internal code CE012006; date 01/31/2020". Line 400
"Statement of Informed Consent: Informed consent was obtained from all subjects involved in the study. "Line 402.
Additionally, the favorable approval report from the ethics committee of the Catholic University of Murcia is attached, as well as a sample informed consent signed by each of the patients participating in the study.
- The study shows a novel method to evaluate the distances from the upper first and second molars to the WALA ridge. Therefore, a comparison to the previous method should provide conclusions to accept it as a better way to have clinical applications. The conclusions of this study give data on the new method, but we do not know if these give us a better and more exact way to obtain a proper diagnose.
REPLY: Studies based on analysis of DICOM files from CBCT may be subject to scatter problems generated in CBCT due to high enamel density, the presence of dental restorations, orthodontic appliances or dental implants [Schulze et al, 2011]. Studies such as that of Sadek et al [Sadek et al, 2014] analyzing posterior alveolar thickness in both the maxilla and mandible using CBCT data inform us that "partial volume averaging can influence spatial resolution. When the voxel size is larger than the object it represents, most often along the margin of an object or at the boundary of two substances of different densities, an average of the densities present is displayed. This can result in lower spatial resolution, as it makes the boundaries between densities more difficult to distinguish accurately." These shortcomings could affect linear measurements on soft tissue. In our study, an analysis method was performed using a DICOM-STL file overlay that had been previously used by [Kim et al, 2016]. This author, performing linear bone and gingival measurements using this DICOM-STL integration made the following statement: "Compared with other studies [Younes et al, 2016], this method was associated with a smaller possibility of errors resulting from impression procedures and the use of bulky ultrasonic instruments....". Regarding the suitability of DICOM-STL overlay, the decision was made to carry it out since it has been found to be a reliable method, as mentioned by Bingshuang Zou et al [Zou et al, 2022] in their study : "this study showed that the integration of maxillary digital models into CBCT scans was clinically reliable". Also, Baan F et al. [Baan et al, 2021] comparing different software stated that DICOM-STL file overlay is a reliable method "The inter- and intra-observer intra-class correlation coefficient showed a high level of agreement between the observers. Both software packages can be used as an accurate intraoral scan fusion tool in CBCT that provides an accurate basis for 3D virtual planning."
Baan F, Bruggink R, Nijsink J, Maal TJJ, Ongkosuwito EM. Fusion of intra-oral scans in cone-beam computed tomography scans. Clin Oral Investig. 2021 Jan;25(1):77-85.
Bingshuang Zou, Jung‑Hoon Kim, So‑Hyun Kim , Tae‑Hyun Choi , Yonsoo Shin , Yoon‑Ah Kook, Nam‑Ki Lee. Accuracy of a surface‑based fusion method when integrating digital models and the cone beam computed tomography scans with metal artifacts. Sci Rep 2022; 12: 8034.
Kim YJ, Park JM, Kim S, et al. New method of assessing the relationship between buccal bone thickness and gingival thickness. J. Periodontal Implant Sci 2016;46(6):372–81
Sadek MM, Sabet NE, Hassan IT. Alveolar bone mapping in subjects with different vertical facial dimensions. Eur J Orthod 2014;37(2):194–201
Schulze R, Heil U, Gross D, Bruellmann DD, Dranischnikow E, Schwanecke U, Schoemer E. Artefacts in CBCT: a review. Dentomaxillofac Radiol. 2011 Jul;40(5):265-73.
Younes F, Eghbali A, Raes M, De Bruyckere T, Cosyn J, De Bruyn H. Relationship between buccal bone and gingival thickness revisited using non-invasive registration methods. Clin Oral Implants Res 2016;27:523-8.
